# Clinical Predictors of Neurogenic Lower Urinary Tract Dysfunction in Persons with Multiple Sclerosis

**DOI:** 10.3390/diagnostics12010191

**Published:** 2022-01-13

**Authors:** Janina Beck, Anke Kirsten Jaekel, Federico Leopoldo Zeller, Michael Kowollik, Ines Kurze, Albert Kaufmann, Wolfgang Feneberg, Anna Brandt, Peter Flachenecker, Thomas Henze, Burkhard Domurath, Paul Schmidt, Will Nelson Vance, Franziska Goldschmidt, Ruth Klara Maria Kirschner-Hermanns, Stephanie C. Knüpfer

**Affiliations:** 1Department for Neuro-Urology, Clinic for Urology, University Hospital Bonn, 53127 Bonn, Germany; Janina.beck@gmx.de (J.B.); Franziska.Goldschmidt@ukbonn.de (F.G.); Ruth.Kirschner-Hermanns@ukbonn.de (R.K.M.K.-H.); stephanie.knuepfer@ukbonn.de (S.C.K.); 2Neuro-Urology, Johanniter Neurological Rehabilitation Center ‘Godeshoehe e.V.’, 53177 Bonn, Germany; f.zeller@godeshoehe.de (F.L.Z.); kowollik@godeshoehe.de (M.K.); 3Center of Spinal Cord Injuries and Diseases, Department for Paraplegiology and Neuro-Urology, 99438 Bad Berka, Germany; ines.kurze@zentralklinik.de; 4Department of Neuro-Urology, Kliniken Maria Hilf GmbH, 41063 Moenchengladbach, Germany; Albert.Kaufmann@mariahilf.de; 5Marianne Strauss Clinic Berg, Therapeutical Center for Patients with Multiple Sclerosis Kempfenhausen GmbH, 82335 Berg, Germany; wolfgang.feneberg@ms-klinik.de; 6Clinic Segeberg, Neurological Center, 23795 Bad Segeberg, Germany; anna.brandt@segebergerkliniken.de; 7Neurological Rehabilitation Center Quellenhof, 75323 Bad Wildbad, Germany; Peter.Flachenecker@Sana.de; 8Neurological Outpatient Practice Dr. Blersch, 93059 Regensburg, Germany; Thomas.Henze@outlook.com; 9Neuro-Urological Center, Clinic Beelitz GmbH, Neurological Rehabilitation Clinic, Beelitz-Heilstätten, 14547 Beelitz, Germany; bdomurath@yahoo.de (B.D.); Vance@kliniken-beelitz.de (W.N.V.); 10Statistical Consulting for Science and Research, Große Seestr. 8, 13086 Berlin, Germany; paul.schmidt.mail@gmail.com

**Keywords:** multiple sclerosis (MS), neuro-urology, neurogenic lower urinary tract dysfunction (NLUTD), Expanded Disability Status Scale (EDSS), post-void residual (PVR), upper urinary tract damage (UUTD), prospective study, bladder diary (BD)

## Abstract

Background: Multiple sclerosis patients often develop neurogenic lower urinary tract dysfunction with a potential risk of upper urinary tract damage. Diagnostic tools are urodynamics, bladder diary, uroflowmetry, and post-void residual, but recommendations for their use are controversial. Objective: We aimed to identify clinical parameters indicative of neurogenic lower urinary tract dysfunction in multiple sclerosis patients. Methods: 207 patients were prospectively assessed independent of the presence of lower urinary tract symptoms. We analyzed Expanded Disability Status Scale scores, uroflowmetry, post-void residual, rate of urinary tract infections, standardized voiding frequency, and voided volume in correlation with urodynamic findings. Results: We found a significant correlation between post-void residual (odds ratio (OR) 4.17, confidence interval (CI) 1.20–22.46), urinary tract infection rate (OR 3.91, CI 1.13–21.0), voided volume (OR 4.53, CI 1.85–11.99), increased standardized voiding frequency (OR 7.40, CI 2.15–39.66), and urodynamic findings indicative of neurogenic lower urinary tract dysfunction. Expanded Disability Status Scale shows no correlation. Those parameters (except post-void residual) are also associated with reduced bladder compliance, as potential risk for kidney damage. Conclusion: Therefore, bladder diary and urinary tract infection rate should be routinely assessed to identify patients who require urodynamics.

## 1. Introduction

Neurogenic lower urinary tract dysfunction (NLUTD) is a common and debilitating manifestation of multiple sclerosis (MS). For many people with MS, urinary symptoms may be the most important socially disabling consequences of the condition [1].

Although urinary symptoms are rare (3–10%) at the first presentation of MS, up to 90% of patients experience neurogenic lower urinary tract symptoms (NLUTS) over the course of their disease [2]. Patients suffer mostly from detrusor overactivity (DO) (65%), hypocontractile detrusor (25%), and detrusor sphincter dyssynergia (DSD) (35%) [3].

The severity of MS is rated using the Expanded Disability Status Scale (EDSS) [4]. This scale assesses the level of disability in a range of functional systems, including the bowel and lower urinary tract, to deliver a total score of 0 to 10. Some studies reported several EDSS threshold levels indicative of NLUTD [5,6,7]. One study found correlations between EDSS ≥ 5.0 and risk factors for upper urinary tract damage (UUTD) [8]. However, a uniform EDSS threshold does not exist.

Furthermore, there is no consensus on the optimal urological management of MS, although several European national panels have published their own guidelines for such patients [2]. These guidelines are nevertheless contradictory [2], and there is no uniform recommendation regarding referral for urodynamic studies (UDS) [8].

We therefore aimed to investigate clinical parameters and EDSS concerning their predictive value for NLUTD and the risk for UUTD in people with MS.

## 2. Materials and Methods

### 2.1. Patients and Assessment

We prospectively included 207 patients with MS originating from 6 clinics specializing in neuro-urology between February 2017 to June 2019.

Inclusion criteria were age > 18 years and written informed consent.

We excluded persons with age < 18 years, non-neurogenic lower urinary tract symptoms (LUTS), pregnancy or breastfeeding, untreated acute lower UTI, or without written informed consent.

All patients completed a two-day bladder diary (BD) to capture data on voided volume per micturition and 24 h voiding frequency.

In addition, we conducted uroflowmetry [9] (including post-void residual (PVR)) and UDS according to ICS standards [10]. Every patient was assessed by EDSS. Information on the occurrence of treated UTIs in the last 6 months was gathered in the patient history.

We investigated correlations between EDSS ≥ 5 and risk factors for UUTD in our cohort. We chose the EDSS threshold of ≥ 5 with regard to the study by Ineichen et al. [8] and defined the risk factors for UUTD as DO combined with DSD or a reduced bladder compliance < 20 mL/cm H_2_O accordingly.

In a further step, we assessed our data to find a threshold value of EDSS regardless of a pre-set threshold which indicates a risk of UUTD or NLUTD.

UDS findings indicative of NLUTD were defined according to current doctrine [9]: first desire to void < 100 mL or strong desire to void < 250 mL or abnormal sensation or bladder capacity < 250 mL or bladder compliance < 20 mL/cm H_2_O or any type of DO or DSD.

Furthermore, we assessed the correlation between different clinical parameters and pathological UDS indicative of NLUTD and potential UUTD. We defined the clinical parameters as follows:
Voided volume (VV)≤250 mL or ≥500 mLUrinary tract infections (UTI) rate>0/6 month24 h standardized voiding frequency (SVF)≤4 or ≥13Post-void residual (PVR)>70 mL and >100 mLUroflowmetry [9]abnormal curve or PVR > 100 mL ormax flow rate < 10 mL/s

The cut-off values of VV were chosen according to a preliminary analysis of our study cohort and according to good urodynamic practice [9]. The thresholds of SVF, UTI, and PVR were defined according to the results of a previous investigation [11]. Voiding frequency was standardized (SVF) to a daily urine outtake of 2000 mL. The following formula was used [11]:$$\mathrm{SVF}=\frac{2000\;\mathrm{mL}}{voided\;volume\;\left(\mathrm{mL}/24h\right)}\;\times \;ø\;\mathrm{voiding}\;\mathrm{frequency}.$$

The cut-off values and diagnostics of uroflowmetry were defined according to good urodynamic practice [9].

In addition, we analyzed the correlation between combinations of our defined clinical parameters and pathological UDS findings defined by Ineichen et al. [8] with regard to the potential risk of UUTD:-Correlation between SVF ≥ 13 + VV ≤ 250 mL and compliance < 20 mL/cm H_2_O.-Correlation between SVF ≥ 13 + PVR > 100 mL and DSD and DO.-Correlation between UTI > 0/6 months + PVR > 100 mL and DSD and DO.

This study was conducted in accordance with the Declaration of Helsinki. All patients gave their written informed consent. Ethical approval (EK 313/13-University Hospital Bonn) was obtained.

### 2.2. Statistical Analysis

For all analyses, R language for statistical computing (Version 3.6.0, R Core Team 2019) was used [12]. A statistical assessment of the threshold EDSS was analyzed with 2 × 2 contingency tables. Clinical parameters correlated to risk factors for NLUTD and UUTD were analyzed with 2 × 2 contingency tables. Fisher’s exact test was used to calculate odds ratios (OR’s) and their 95% confidence limits. To assess the prognostic quality of the selected predictors, standard performance measures for binary classifiers were used, namely sensitivity (Sens), specificity (Spec), positive (PPV), and negative predictive values (NPV).

## 3. Results

### 3.1. Patient Characteristics, Clinical Parameters, and UDS Findings

We assessed 207 patients, of whom 69.1% (141) were female and 30.9% (63) were male. Patient and disease characteristics are summarized in Table 1.

A summary of the collected parameters from BD, uroflowmetry, and UDS, as well as EDSS, is shown in Table 2.

In our study, 83% (172) of patients with MS had a pathological UDS outcome indicative of NLUTD; 20% (39) showed risk factors for UUTD.

Furthermore, 79% (84/106) of asymptomatic patients (no urinary symptoms) showed abnormal UDS findings indicative of NLUTD, whereas 13% (13/99) of asymptomatic patients showed risk factors for UUTD in their UDS findings. A summary of the presence of LUTS and UDS findings are shown in Table 3.

### 3.2. EDSS Threshold

The cohort was analyzed with regard to risk factors for UUTD in two patient groups: EDSS < 5 vs. EDSS ≥ 5.

The contingency table showed that there is no obvious difference between reduced compliance with an EDSS < 5 vs. EDSS ≥ 5 (16.7% vs. 12.7%; Table 4). A similar result is obtained for the patients with DO and DSD (9.3% vs. 14.1%; Table 4).

The data showed no significant differences between the two groups of EDSS (<5 and ≥5) concerning the risk of reduced compliance (*p* = 0.53) or DO and DSD (*p* = 0.34). The predictive power (Sens, Spec, PPV, NPV) of the EDSS threshold value of 5 is limited (Table 5).

Data were analyzed to determine a threshold of EDSS that indicated NLUTD or potential UUTD (Figure 1). No increasing or decreasing trend was observed. The AUC-performance measures of the three parameters on the *y*-axis did not exceed 0.56.

### 3.3. Correlations between Clinical Parameters and UDS Findings Indicative of NLUTD or Potential UUTD

The following clinical parameters had significant correlations with UDS findings indicative of NLUTD: PVR (*p* = 0.02/0.00), SVF ≥ 13/24 h (*p* = 0.00), VV ≤ 250 mL (*p* = 0.00) and ≥ 500 mL (*p* = 0.02), and UTI > 0/6 months (*p* = 0.03) (Table 6).

VV ≥ 500 mL indicates a lower risk of NLUTD with OR 0.3 (CI 0.11–0.87) and *p*-value 0.02, but the sensitivity is only 10%. SVF ≤ 4/24 h shows a significant effect on NLUTD with a *p*-value of 0.03 in two patients.

Significant correlations were found between reduced compliance and the following clinical parameters: VV ≤ 250 mL (*p* = 0.01), SVF ≥ 13/24 h (*p* = 0.00) and UTI > 0/6 months (*p* = 0.04); the OR was > 1.0. VV ≤ 250 mL showed the best sensitivity with 83%, but a low PPV (19%). SVF ≥ 13/24 h and UTI > 0/6 months showed a sensitivity of 66% and 43% (Table 6).

The correlation of clinical parameters with DO and DSD did not lead to a significant result.

### 3.4. Influence of Combined Clinical Parameters on UDS Findings Indicative of Potential UUTD

The combination of SVF ≥ 13/24 h and VV ≤ 250 mL showed a significant correlation with compliance < 20 mL/cm H_2_O (OR 3.36, CI 1.23–9.06, *p* = 0.009). However, our data indicated a low sensitivity (0.46) and PPV (0.27).

A combined assessment of SVF ≥ 13/24 h and PVR > 100 mL with DO and DSD yielded no statistically significant evidence (*p* = 0.413).

A potential correlation was also observed between the combination of UTI > 0/6 months and PVR > 100 mL with DO and DSD. This combination was not significant (*p* = 1.00).

## 4. Discussion

Several recommendations and consensus documents on the management of urinary disorders in multiple sclerosis patients propose that EDSS is a decision-making factor for further urological investigations [5,8,13,14,15]. However, no global threshold value of EDSS exists [2]. Furthermore, the clinical value with regard to storage and voiding symptoms is still unclear [16].

Our data demonstrate no reliable correlation between EDSS ≥ 5 and the risk of UUTD. Furthermore, we found no other threshold value of EDSS that indicates NLUTD or the risk of UUTD. In contrast to our data, previous studies have demonstrated a significant association between EDSS and abnormal UDS findings [5,6,7,8]. However, only one study [8] investigated the correlation between EDSS and risk factors of UUTD, and it included only patients with lower urinary tract symptoms (LUTS). In contrast, we assessed a cohort that was not divided into LUTS and non-LUTS cases. This may have affected the detailed composition of the EDSS and thus biased the results. Ineichen et al. [8] stated that most patients with higher EDSS will have LUTS. Despite this, due to the impact of eight functional systems (pyramidal, cerebellar, brainstem, sensory, bowel and bladder, visual, cerebral, other) on the EDSS, patients with a similar EDSS are not necessarily a homogenous group in terms of LUTS. Wiedemann et al. [5] assessed only patients with LUTS and had a mean EDSS similar to ours: 4.5 (±2.3). Based on their findings, they recommended a UDS for all patients with EDSS ≥ 6.5 [5]. In contrast, Nakipoglu et al. [17] investigated a cohort with and without LUTS. They did not establish a relationship between disease characteristics and urodynamic findings. Their mean EDSS was 5.1 (±2.2) and similar to the cohort with LUTS of Ineichen et al. [8]. A high diversity regarding correlations between EDSS, LUTS, and urodynamic findings also exists in other studies [6,7,18]. The inconsistent definition of pathological UDS findings and the bias of UDS interpretation hamper a comparison of the various studies. Nevertheless, the lack of a standard EDSS threshold is reflected in the different guidelines/recommendations. Italian [15] and Spanish [14] consensus documents recommend further neuro-urological examination based on an EDSS of 3, whereas a French algorithm emphasized an EDSS ≥ 6 as a red flag for neuro-urological consultation [13]. In contrast, a recent multinational consensus statement [16] presented EDSS as a useful tool for measuring the progression of neurological disability, but in terms of assessing LUTS in persons with MS the consensus group recommends LUT-specific questionnaires instead of EDSS [16].

NLUTS are common symptoms among persons with MS [2] and can significantly impair quality of life [3]. Furthermore, NLUTD is the main reason for morbidity and hospitalization [6]. Nevertheless, published data suggest an under-diagnosis of NLUTD (up to 48%) and non-optimized or under-treatment in persons with MS [14]. Previous studies have highlighted that a significant proportion of asymptomatic patients with MS have NLUTD [18,19], and the absence of symptoms is poorly reflected by UDS [3]. Focusing on asymptomatic patients, the present study revealed UDS results indicative of NLUTD in 79% of asymptomatic patients, whereas 13% of asymptomatic patients showed risk factors for UUTD. A similar result was described by Bemelmans et al. [18], who revealed abnormal UDS results in 52% of patients without any urinary complaints. They saw a hyposensitive bladder as a reason for this number of asymptomatic patients. These results are in line with Nakipoglu et al., who could not find a significant correlation between urinary symptoms and urodynamic findings [17]. However, this issue is reflected in different guidelines that recommend more in-depth examinations for asymptomatic patients [14,15,19], even though the algorithms recommending UDS are different. In contrast, the UK consensus recommends bladder management corresponding to LUTS [20]: UDS is recommended only in the event of surgical treatment or if conservative treatment fails. The British expert panel pointed out that UDS results are unlikely to influence management procedures and UUTD is rare in persons with MS [20]. Similarly, the Turkish consensus report recommends invasive UDS only in the event of UUTD and/or failure of conservative treatment [21]. On the other hand, other guidelines state that adequate treatment can reduce the risk of UUTD or even prevent it, and it is essential to detect at-risk patients as early as possible [8,14]; therefore, there is a need for additional parameters indicating NLUTD regardless of the presence of LUTS. Our study showed that additional clinical parameters were useful for identifying patients who required further neuro-urological assessment. In addition to the well-known parameters PVR and rate of UTI, we found increased SVF to be indicative of NLUTD and UUTD. SVF was first assessed by Domurath et al. [11] as part of a newly evaluated algorithm in the neuro-urological assessment of persons with MS. A significant correlation between DO and SVF was found with a cut off ≥ 13/24 h, 95.5% of the patients showed abnormal UDS results [11]. We assessed increased SVF as a single parameter and obtained significant correlations to UDS findings indicative of NLUTD. Furthermore, our data showed a significant correlation with reduced bladder compliance, which is considered a risk factor for UUTD [8].

According to our data, another novel clinical parameter in terms of NLUTD is VV ≤ 250 mL. It proved to be indicative of NLUTD and the risk of UUTD and had the best prognostic performance of the tested parameters. VV has been shown to be a useful measure of efficacy regarding the medical treatment of overactive bladder (OAB) [22]. Van Brummen et al. [23] demonstrated an association between frequent symptoms of OAB and lower maximum VV. To the best of our knowledge, our study is the first to assess VV with regard to abnormal UDS findings in persons with MS and no cut-off of voided volume in relation to NLUTD exists; therefore, we used the cut-off value of 250 mL in accordance with our BD data. Even though the use of bladder diaries is recommended in several guidelines [13,15,16,19], the recommended criteria for further investigations differ considerably [15,19]. None of the recommendations are based on the details of a BD, and all guidelines [13,15,16,19] point out that a BD should be used if patients report LUTS.

For this reason, we must emphasize that urological symptoms in our study cohort poorly reflect abnormalities in SVF or VV. 52% of the asymptomatic patients with NLUTD had bladder diaries with a VV ≤ 250 mL. 35% of the asymptomatic patients with NLUTD showed an SVF ≥ 13/24 h. This underlines the need for an objective tool for determining abnormalities in voiding issues. A bladder diary can be this tool as it is a simple, non-invasive, and cost-efficient instrument for objectifying symptoms and disorders. Furthermore, the asymptomatic patients included almost 22% with UTI > 0/6 months; therefore, we recommend the routine use of a bladder diary and an assessment of UTIs in persons with MS. Those with VV ≤ 250 mL, SVF ≥ 13/24 h, or UTI > 0/6 months should soon undergo UDS.

It must be noted that risk factors for UUTD were adopted from spina-bifida and spinal-cord-injury patients, who have a higher incidence of kidney damage than persons with MS [24]. Furthermore, it must be borne in mind that UDS in asymptomatic healthy individuals have generated variable results [25]. Finally, data were collected by highly specialized neuro-urological departments, and thus the recorded baseline characteristics of patients may deviate from those in less specialist settings.

## 5. Conclusions

Persons with MS should be examined for NLUTD regardless of LUTS. Our study revealed that voided volume ≤ 250 mL, voiding frequency ≥ 13/24 h, and UTI > 0/6 months are clinical parameters indicative of NLUTD and the potential risk of UUTD in persons with MS. To determine these parameters, BD and an assessment of UTIs are mandatory for every patient with MS. Affected patients should soon undergo UDS.

## Figures and Tables

**Figure 1 diagnostics-12-00191-f001:**
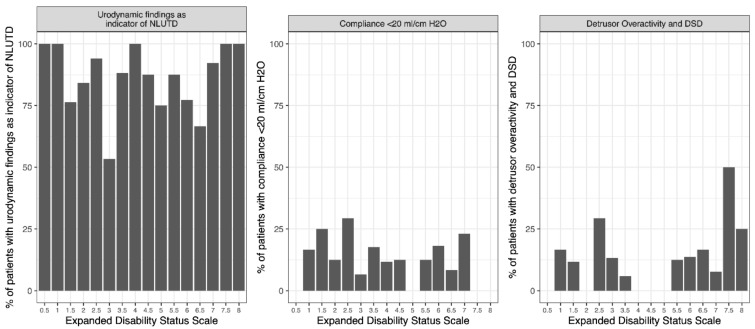
Analysis of Expanded Disability Status Scale threshold. NLUTD, neurogenic lower urinary tract dysfunction; DSD, detrusor sphincter dyssynergia.

**Table 1 diagnostics-12-00191-t001:** Patient and disease characteristics.

	Mean (SD)	Median (25–75%)	Min–Max	Missing % (*n*)
Age of patients in years	49.2 (10.7)	49 (41–55)	19–75	1.4% (3)
Age of MS onset in years	35.5 (11)	34 (28–42)	14–71	2.9% (6)
Disease duration in years	13.7 (9.5)	13 (6–20)	0–46	2.9% (6)
MS Type	% (*n*)			
PPMS	10.9% (22)			2.4% (5)
RRMS	46.5% (94)			
SPMS	42.6% (86)			

SD, standard deviation; MS, multiple sclerosis; PPMS, primary progressive MS; RRMS, relapsing remitting MS; SPMS, secondary progressive MS.

**Table 2 diagnostics-12-00191-t002:** Summary of the parameters of BD, uroflowmetry, UDS, and EDSS.

		% (*n*)	Mean (SD)	Median (25–75%)	Min–Max	Missing % (*n*)
BD	Daily fluid intake [mL]		1908.7 (619.1)	1850 (1500–2200)	500–5000	2.9% (6)
	Daily urine outtake [mL]		1705.3 (689.7)	1600 (1200–2158)	555–3950	19.3% (40)
	Average VV [mL]		239 (122.6)	215 (150–300)	50–925	18.4% (38)
	Average VF at day		8.2 (3.8)	7 (6–10)	2–30	8.7% (18)
	Average VF at night		1.8 (1.9)	1 (1–2)	0–14	7.7% (16)
UTI	Per 6 month					
	0	74.2% (147)				4.3% (9)
	1	9.6% (19)				
	2	5.6% (11)				
	3	6.6% (13)				
	>3	4% (8)				
EDSS			4.1 (2)	4 (2.5–6)	0.5–8	8.7% (18)
UF	Qmax [mL/s]		1 (12)	16.7 (11–23)	0–68.1	19.3% (40)
	VV [mL]		260.8 (212.3)	209 (120–336.5)	0–1300	14% (29)
	PVR [mL]		80.5 (101.7)	42.5 (12.8–113.2)	0–580	11.1% (23)
	Abnormal curve	55.7% (98)				15% (31)
UDS	First desire to void [mL]		207.5 (126.9)	177 (112–286)	8–710	6.8% (14)
	Strong desire to void [mL]		302.3 (141.5)	286 (202–385)	33–828	14.5% (30)
	Compliance [mL/cm H_2_O]		63.4 (60.1)	52 (25–83)	0–453.4	3.4% (7)
	DO	40.1% (81)				2.4% (5)
	DSD	31.2% (59)				8.7% (18)
	Max. BC [mL]		405.8 (158)	401 (299–495)	80–1000	3.4% (7)

BD, bladder diary; SD, standard deviation; UTI, urinary tract infections; EDSS, Expanded Disability Status Scale; UF, uroflowmetry; VV, voided volume; VF, voiding frequency; PVR, post-void residual; Qmax, maximum uroflow; UDS, urodynamic study; DO, detrusor overactivity; DSD, detrusor sphincter dyssynergia; max. BC, maximum cystometric bladder capacity.

**Table 3 diagnostics-12-00191-t003:** The presence of LUTS and UDS findings indicative of NLUTD and potential UUTD.

Symptomatology with Regard to NLUTD			
		Abnormal UDS	
	*n*	%	*n*
Symptomatic	101	87	88
Asymptomatic	106	79	84
**Symptomatology with Regard to Risk of UUTD**			
		Abnormal UDS	
	*n*	%	*n*
Symptomatic	101	26	26
Asymptomatic	99	13	13

LUTS, lower urinary tract symptoms; UDS, urodynamic studies; NLUTD, neurogenic lower urinary tract dysfunction; UUTD, upper urinary tract damage.

**Table 4 diagnostics-12-00191-t004:** Frequency distribution of EDSS and potential risk factors for UUTD.

EDSS	Compliance < 20 mL/cm H_2_O		DO and DSD	
	No	Yes	No	Yes
<5	83.3% (95)	16.7% (19)	90.7% (107)	9.3% (11)
≥5	87.3% (62)	12.7% (9)	85.9% (61)	14.1% (10)

DO, Detrusor overactivity; DSD, detrusor sphincter dyssynergia; EDSS, Expanded Disability Status Scale.

**Table 5 diagnostics-12-00191-t005:** *p*-values and predictive parameters of EDSS and risk factors for UUTD.

	Odds Ratio	* p * -Value	Sens	Spec	PPV	NPV
Compliance < 20 mL/cm H_2_O	0.73 (0.27–1.82)	0.53	0.32	0.61	0.13	0.83
DO and DSD	1.59 (0.57–4.39)	0.34	0.48	0.64	0.14	0.91

UUTD, upper urinary tract damage; DO, detrusor overactivity; DSD, detrusor sphincter dyssynergia; Sens, sensitivity; Spec, specificity; PPV, positive predictive value; NPV, negative predictive value; significance level of *p*-value < 0.05.

**Table 6 diagnostics-12-00191-t006:** Relationship between clinical parameters and UDS indicative of NLUTD and between clinical parameters and reduced compliance.

Urodynamic Findings Indicative of NLUTD						
	Odds Ratio	*p*-Value	Sens	Spec	PPV	NPV
SVF ≤ 4/24 h	0 (0–1.06)	0.03	0.00	0.94	0.00	0.16
SVF ≥ 13/24 h	7.4 (2.15–39.66)	0.00	0.44	0.90	0.96	0.25
VV ≤ 250 mL	4.53 (1.85–11.99)	0.00	0.64	0.72	0.91	0.31
VV ≥ 500 mL	0.3 (0.11–0.87)	0.02	0.10	0.72	0.62	0.15
Uroflowmetry	1.91 (0.75–5.02)	0.20	0.54	0.62	0.81	0.31
PVR > 70 mL	6.43 (1.87–34.4)	0.00	0.40	0.91	0.95	0.24
PVR > 100 mL	4.17 (1.20–22.46)	0.02	0.30	0.91	0.94	0.21
UTI > 0/6 months	3.91 (1.13–21.00)	0.03	0.29	0.91	0.94	0.20
**Compliance < 20 mL/cm H_2_O**						
	**Odds Ratio**	** *p* ** **-Value**	**Sens**	**Spec**	**PPV**	**NPV**
SVF ≤ 4/24 h	0 (0–28.03)	1.00	0.00	0.99	0.00	0.84
SVF ≥ 13/24 h	3.7 (1.51–9.61)	0.00	0.66	0.66	0.27	0.91
VV ≤ 250 mL	3.91 (1.22–16.56)	0.01	0.83	0.45	0.19	0.94
VV ≥ 500 mL	0.59 (0.06–2.71)	0.74	0.09	0.86	0.09	0.86
Uroflowmetry	0.25 (0.02–1.39)	0.09	0.22	0.46	0.03	0.88
PVR > 70 mL	0.75 (0.27–1.94)	0.66	0.30	0.64	0.13	0.84
PVR > 100 mL	0.76 (0.23–2.11)	0.64	0.22	0.73	0.12	0.84
UTI > 0/6 months	2.52 (1.03–6.10)	0.04	0.43	0.77	0.25	0.88

## Data Availability

The data presented in this study are available on request from the corresponding author. The data are not publicly available due to privacy.

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
