# Peer review of "Clinical Predictors of Neurogenic Lower Urinary Tract Dysfunction in Persons with Multiple Sclerosis"

_diagnostics, 2022, doi:10.3390/diagnostics12010191_

Round 1
Reviewer 1 Report
This manuscript shows the clinical parameters predicting progression of lower urinary tract dysfunction in patients with multiple sclerosis. The purpose of this study may be useful and contribute to treatment of lower urinary tract dysfunction with MS. However, there are several concerns related to the study design and discussion.
- Please summarize the baseline characteristics of all 207 MS patients in new Table (age, sex, BMI, diseases, etc).
- In addition, please summarize all raw data of bladder diary, uroflowmetry, EDSS etc in details.
- Table 1 is not necessary. Please replace it to patient characteristics (new Table 1).
- Table 2 is not necessary. Please replace it to raw data of objective parameters (new Table 2).
- Please show why the authors chose cut-off of EDSS as 5.
- Also, why 250, 500 ml of voided volume? New table of data including bladder diary, uroflowmetry, EDSS could help to lead to cut-off level.
- Other your cut-off values are same. Please define these.
Reviewer 2 Report
The authors aimed to identify clinical parameters indicative of neurogenic lower urinary tract dysfunction in multiple sclerosis patients.
As they stated neither global threshold value of EDSS exists, nor the clinical value with regard to storage and voiding symptoms is still unclear.
They found that bladder diary and urinary tract infection rate should be routinely assessed to identify patients who require urodynamics.
From this point of view, this report is interesting.
This paper has the potential to be widely applied in the future.
Round 2
Reviewer 1 Report
The authors have revised accordingly.